# Esophageal Magnetic Compression Anastomosis in Esophageal Atresia Repair: A PRISMA-Compliant Systematic Review and Comparison with a Novel Approach

**DOI:** 10.3390/children9081113

**Published:** 2022-07-25

**Authors:** Anne-Sophie Holler, Tatjana Tamara König, Caressa Chen, Michael R. Harrison, Oliver J. Muensterer

**Affiliations:** 1Department of Pediatric Surgery, Dr. von Hauner Children’s Hospital, University Hospital, LMU Munich, 55131 Mainz, Germany; oliver.muensterer@med.uni-muenchen.de; 2Department of Pediatric Surgery, Universitätsmedizin, Johannes-Gutenberg University, 55131 Mainz, Germany; tatjana.koenig@unimedizin-mainz.de; 3Department of Surgery and Bioengineering & Therapeutic Sciences, University of California, San Francisco, CA 94102, USA; caressa.chen@ucsf.edu; 4Division of Pediatric Surgery, Department of Surgery, University of California, San Francisco, CA 94102, USA; michael.harrison@ucsf.edu

**Keywords:** esophageal atresia, magnet, anastomosis, long-gap

## Abstract

The use of magnet compression to endoscopically create an esophageal anastomosis is an intriguing approach to esophageal atresia repair, but published cases with an existing available device have demonstrated mixed success. One major shortcoming has been the formation of subsequent severe, recalcitrant strictures after primary repair. To address the limitations of the existing device, we recently introduced and reported success with specially designed bi-radial magnets that exhibit a novel geometry and unique tissue compression profile. The aim of this study is to compare the outcomes using our novel device (novel group, NG) with those of previous reports which utilized the historical device (historic group, HG) in a PRISMA-compliant systematic review. Seven studies were eligible for further analysis. Additionally, one of our previously unreported cases was included in the analysis. Esophageal pouch approximation prior to primary repair was performed more frequently in the NG than in the HG (100% NG vs. 21% HG; *p* = 0.003). There was no difference in the overall postoperative appearance of postoperative stricture (95% HG vs. 100% NG; *p* = 0.64). The number of postoperative dilatations trended lower in the NG (mean 4.25 NG vs. 9.5 HG; *p* = 0.051). In summary, magnetic compression anastomosis adds a new promising treatment option for patients with complex esophageal atresia. Prior approximation of pouches and a novel magnet design have the potential to lower the rate of stricture formation.

## 1. Introduction

Esophageal atresia (EA) repair can be performed either via thoracotomy or by a minimally invasive approach. However, the thoracoscopic procedure is one of the most challenging procedures in pediatric surgery and, consequently, has not been widely adopted [1,2]. The presence of long-gap EA adds an additional challenge to either open or thoracoscopic repair. Many different approaches to address long-gap EA have been described, including lengthening and esophageal replacement procedures [3,4]. Preserving the native esophagus is generally considered a priority for patients, as esophageal replacement is associated with high morbidity and mortality [5].

In 1976, Hendren first described the use of an electromagnetic field in combination with metal bougies to approximate the esophageal ends in long-gap EA [6]. Much later, in 2009, Zaritzky et al. published a case series of patients with EA who underwent placement of magnets in the esophageal pouches to achieve lengthening and subsequent anastomosis by tissue compression [7]. Unfortunately, this procedure was associated with a high number of anastomotic strictures requiring repetitive dilatations [8]. Since then, a small number of case reports and case series have been published using these original, and other, magnets [9,10,11,12,13].

Our research has focused on magnet size and geometry, with the aim of optimizing the devices to create a robust anastomosis and reduce post-interventional stricture rates [14,15,16]. The newly designed magnets were validated in an experimental animal model, which demonstrated highly favorable short- and long-term outcomes [14,16]. Furthermore, the first in-human experience was published in 2021 with promising results [15].

The aim of this study was to perform a systematic review of the literature which compares outcomes of magnetic esophageal compression anastomosis using our novel device to those using historic magnets. We also report a previously unpublished case using our new device and include it in the analysis.

## 2. Materials and Methods

### 2.1. Systematic Review

The systematic review was prepared according to the Preferred Reporting Items for Systematic Reviews and Meta-Analyses statement (PRISMA) [17]. A literature search via Medline, Embase, Web of Science and the Cochrane Library for Reviews was conducted in November 2021 using a defined search strategy (Table 1). The search terms “magnets” AND “anastomosis” AND “esophagus” were used. Furthermore, the reference list of eligible studies and congress issues from 2021 were searched. We defined the following inclusion and exclusion criteria: Inclusion criteria: (1) pediatric population, (2) esophageal anastomosis, (3) use of magnets. Exclusion criteria: (1) Experimental studies or animal studies, (2) adult patients, (3) anastomosis on other parts of the gastrointestinal tract, (4) treatment of esophageal stenosis (Table 1). Abstracts were screened for eligibility (PRISMA-Flowchart, Figure 1).

Data was extracted according to a standardized protocol including information on demographic, technical, and outcome parameters. Complications were divided into early (leakage) or late (stenosis) complications.

### 2.2. Subgroup Analysis

Eligible studies were divided into two subgroups: a historic group (HG) and patients treated with our novel, specially designed magnets (novel group, NG). In addition to 3 previously published cases [15], another case was added and described in the case report section.

We used the newly designed magnets (*Connect-EA*, University of California San Francisco Surgical Innovations, San Francisco, CA, USA) with two 8 mm-diameter magnetic anchors comprised of a neodymium–iron–boron magnetic core encapsulated by biocompatible plating. The new shape is characterized by a convex-concave geometry, which leads to central tissue necrosis and allows peripheral mucosal healing and bridging (Figure 2).

### 2.3. Statistical Analysis

Continuous data was represented as means and standard deviation. We used percentage frequency to express categorial data. Data was compared using the Mann–Whitney-*U* test for continuous data and the Chi-Quadrat test for categorial data. The null hypothesis was rejected with *p*-values < 0.05. Data analysis was conducted using SPSS software (IBM SPSS^®^ Statistics 26).

## 3. Results

### 3.1. Systematic Review

In total, 96 articles published from 2009 through 2021 were identified. Additionally, one article from our group, one cross-reference article, and one congress abstract were included. After removing duplicates, 74 abstracts were screened for inclusion and exclusion criteria. All remaining articles were screened as full texts. Finally, 10 articles—including case reports and congress issues—were eligible for the systematic review (Figure 1). Seven articles were included in the subgroup analysis. The overall level of evidence score ranged from IV to V; all included studies were retrospective case reports or case series with a maximum of 13 patients.

The first small case series reported in 2009 by Zaritzky et al. included five patients from Argentina [18]. Later, in 2014, the same group published a series of nine patients, which included the previously reported five patients [7]. Therefore, data from the 2009 paper was excluded from the subgroup analysis. Furthermore, the same group analyzed long-term follow-up data of 13 patients in 2019 [8]. According to the authors, this analysis also included seven patients from former reports. Overall, this analysis was more detailed and added information on long-term follow-up. For this reason, we decided to exclude the publication by Zaritzky et al. 2014 [7] from the subgroup analysis as well. The data of Conforti et al. was excluded due to lack of long-term follow-up data. Wolfe et al. reported three cases of magnetic compression anastomosis [12]; in addition to one successful magnetic anastomosis, one attempt was unsuccessful, and one patient was treated for stenosis. Therefore, only one case from this series was included in the final analysis.

Overall, data from 23 patients were ultimately analyzed. Demographic data and outcome parameters are presented in Table 2 and Table 3. Information about gap distance was non-uniform—either in centimeters or number of vertebral bodies—and was measured at different ages. Therefore, this parameter could not be included in the further analysis.

### 3.2. Subgroup Analysis

There was no significant difference between the underlying type of EA. In the patients included in our study, Gross Type A EA was the most common (NG 50%, HG 78.8%, *p* = 0.488; Table 4).

Groups differed regarding the interventions that were performed before establishing the magnetic anastomosis. In both groups, gastrostomy was placed in nearly all cases (NG 100%, HG 94.7%, *p* = 0.639). Prior to magnetic compression anastomosis, fistula ligation was performed in patients with Gross Types B or C EA (NG 50%, HG 15.8%, *p* = 0.132). In the NG, significantly more patients underwent prior approximation of the esophageal pouches compared to the HG (NG 100%, HG 21.1%, *p* = 0.003).

The mean age at the time of intervention was significantly lower in the NG (NG mean 2.4 ± 0.8, HG mean 4.63 ± 1.71, *p* = 0.009). There was no statistical difference in time to patency of the anastomosis (NG mean 9.50 ± 2.89, HG mean 8.37 ± 7.4, *p* = 0.186).

Leakage was present in one case in the HG and did not occur in the NG (*p* = 0.639). In both groups, the most common complication was the development of an anastomotic stricture. The overall presence of a stricture did not differ significantly between groups (NG 100%, HG 94.7%, *p* = 0.639). However, the mean number of dilatations trended lower in the NG, without reaching statistical significance (NG mean 4.25 ± 0.50, HG mean: 9.50 ± 6.41, *p* = 0.051). Stent placement for stricture treatment was performed in six patients (35.5%) in the HG whereas stent placement was not required in the NG (*p* = 0.160). Two patients in the HG underwent surgery for recalcitrant stricture (NG none, HG 10.5%, *p* = 0.497), while no patients in the NG required additional procedures.

Mean time of follow up was 1.16 ± 0.33 years in the NG and 6.89 ± 6.24 years in the HG (*p* = 0.066). The native esophagus was successfully preserved in all children, with no need for an intestinal or gastric interposition.

### 3.3. Case Report

A 1965 g twin girl was born at 36 weeks’ gestation with Gross Type A EA. She was transferred to our tertiary center, where further associated anomalies were ruled out. Gastrostomy was performed on her second day of life, with the intention of performing delayed primary repair. At 6 weeks of age, a bronchoscopy and contrast gap study were performed, demonstrating Type A long-gap EA, with a gap of five vertebral bodies (Figure 3a). Subsequently, she underwent thoracoscopic mobilization and approximation of her esophageal pouches with slip-knot sutures (Figure 3b). After 5 weeks, when the esophageal tension was thought to have subsided, magnets were endoscopically placed in the upper and lower pouches and mated under fluoroscopic visualization (Figure 3c). At the end of the procedure, a Replogle tube was reinserted. Postoperatively, the position and migration of the magnets was monitored via serial X-rays. On postoperative day 5, the magnets started tilting, indicating progressive detachment from the esophageal wall. The patient was able to swallow her saliva on postoperative day 7 and subsequently started oral feeding. The mated magnet pair was spontaneously excreted in the stool on postoperative day 11. Contrast esophagography was performed, which indicated a newly formed patent anastomosis without concern of leaks (Figure 3d). After both interventions (approximation and magnet placement), enteral feeds via gastrostomy were commenced on postoperative day 1. At the time of discharge, the patient was tolerating full oral feeds. Four months after the esophageal magnetic compression anastomosis, the patient developed swallowing problems and a contrast esophagography revealed anastomotic stenosis, which responded well to subsequent dilatations. To date, a total of four dilatations have been needed. The gastrostomy was closed at 11 months of age. On last follow-up at 13 months of age, the child was on a full oral age-appropriate diet and showed normal age-appropriate weight gain.

## 4. Discussion

EA is a rare malformation and magnetic compression anastomosis is still an experimental approach for achieving esophageal anastomosis. Therefore, data on magnetic compression anastomosis is scarce and the level of evidence is restricted to retrospective reports. To date, the procedure has been mainly applied as compassionate care therapy in selected complicated cases of long-gap EA, failed primary repair, and/or in patients with significant medical comorbidities.

The major difference between the indications for use of the historic and novel group was that in the HG, magnets were not only used for esophageal anastomosis, but also esophageal lengthening [7,8,18]. Zaritzky et al. defined a maximum gap between the esophageal pouches of 3–4 cm as acceptable for the use of the magnets. Esophageal tension and long-gap EA are independent, well-established risk factors for stricture formation [20,21,22]. In the NG, thoracoscopic esophageal lengthening procedures were performed prior to the magnetic compression anastomosis, separating a tension-creating procedure for lengthening from the procedure for esophageal anastomosis creation. It is important to understand that magnets should not be used for approximation, but only for anastomosis. Therefore, performing a thoracoscopic approximation before placing the magnets is part of the comprehensive treatment strategy in the NG.

Our systematic review demonstrated a trend toward fewer postoperative dilatations in the NG, despite it including fewer patients. Our hypothesis is that less anastomotic tension reduces the rate of anastomotic strictures and, consequently, the number of dilatations, and that the novel curvature and geometry allows for better bridging of the mucosa. Nevertheless, almost all patients in both groups developed some esophageal stenosis. However, even though the difference did not reach statistical significance due to low patient numbers so far, patients in the NG showed a strong trend toward a lower number of dilatations, suggesting a less severe stricture.

The mean follow-up period was longer in the HG, without reaching statistical significance. However, dilatation frequency in the NG decreased with time and a stable state was reached, with dilatations only required in the initial months post-repair.

The majority of patients treated in our series were born with Gross Type A and were characterized as having long-gap EA. The definition of long-gap EA varies, depending on the reference. In general, EA repair in long-gap EA is challenging and associated with high numbers of anastomotic strictures [21,22]. Bagolan et al. retrospectively analyzed 57 patients with long-gap EA. In their cohort, a mean of 4.5 dilatations were needed, and in 14% of cases, retrievable stents were placed [23]. A recently published single center study from Australia retrospectively reviewed 247 patients who underwent EA repair. In this cohort, over 60% of cases (including all types of EA) required postoperative stricture dilatations. In these cases, a mean of four (interquartile range two to eight) dilatations per patient were needed [24]. This is consistent with data from Germany that showed stricture rates of 57% irrespective of the underlying type of atresia [25]. Long-gap EA and pure EA (Gross Type A), as included in our study, are considered individual risk factors for developing refractory stricture [26]. Although the low numbers preclude a statistical analysis, it is important to note that within the NG, no stents or additional operative interventions were necessary—while in the HG, over one third of cases were treated with a stent, and 10% underwent stricture resection and reanastomosis.

The main advantage of endoluminal magnetic compression anastomosis is an endoscopic approach with shorter operative times [15]. With regard to the negative side effects of anesthesia in neonates and the discussed impaired neurodevelopmental outcomes, shorter operative times are especially beneficial in patients with comorbidities [27]. Endoscopic magnet placement in the NG took around 30 min in all cases, while conventional open EA repair with a handsewn anastomosis takes 202 min [28].

According to a recent study, less than one hour of general anesthesia does not alter neurodevelopmental outcomes [27]. Furthermore, our analysis showed a very low leak rate (4.3%). In the NG, there were no leaks; this is consistent with previous animal studies, which demonstrated that the new magnets create a watertight and robust anastomosis [16].

Of the included reports, only one case of magnetic compression anastomosis was unsuccessful [11].

The limitations of the statistical analysis are the low level of evidence in the included studies, with small numbers of patients treated and differing group sizes. Unfortunately, we only have limited information on the gap length in the HG; this also makes the groups difficult to compare. We acknowledge that the study groups are heterogeneous, and that the patients in the HG came from different studies. This could represent a potential source of bias. Furthermore, the group sizes differed; the NG had just four cases whereas the HG included 19 cases. A multicenter, prospective study is currently in preparation to evaluate the potential benefits of magnetic compression anastomosis using the novel magnet design using robust data.

## 5. Conclusions

Esophageal magnetic compression anastomosis is a safe and technically simple procedure that represents an alternative treatment method for patients with complex or long-gap EA, who are at risk for poor outcomes. Advantages of the procedure include the robust anastomosis formation, low leakage rate, and preservation of the native esophagus. High stricture rates are reported in all studies, but the majority responded well to repetitive dilatations. Our new developments in magnet design and geometry, in combination with prior approximation of pouches, have the potential to lower the rate of postoperative dilatations and interventions in EA repair.

## Figures and Tables

**Figure 1 children-09-01113-f001:**
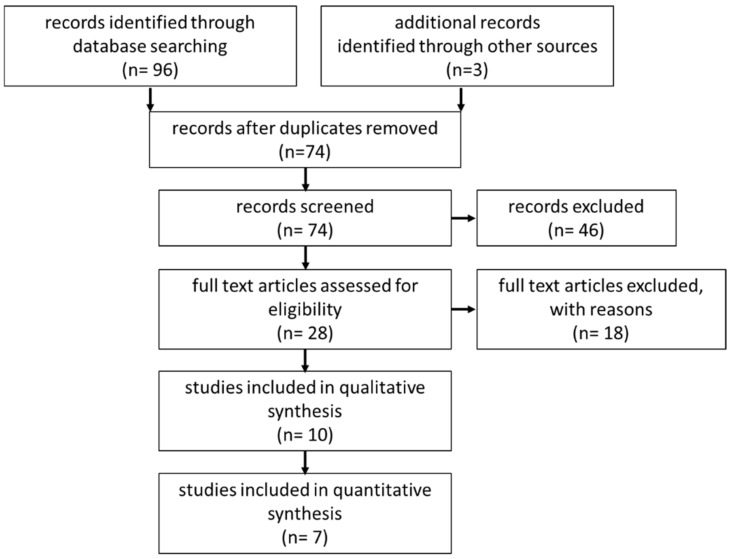
PRISMA-Flowchart of workflow in the systematic review.

**Figure 2 children-09-01113-f002:**
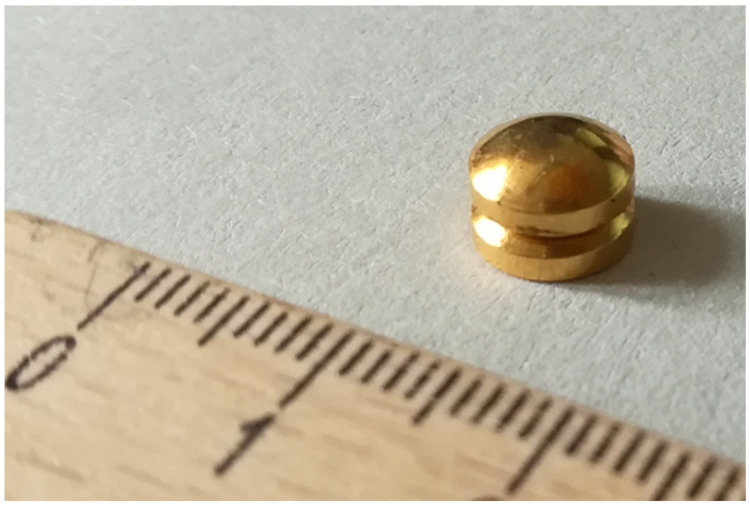
Pair of the newly designed magnets with convex-concave geometry.

**Figure 3 children-09-01113-f003:**
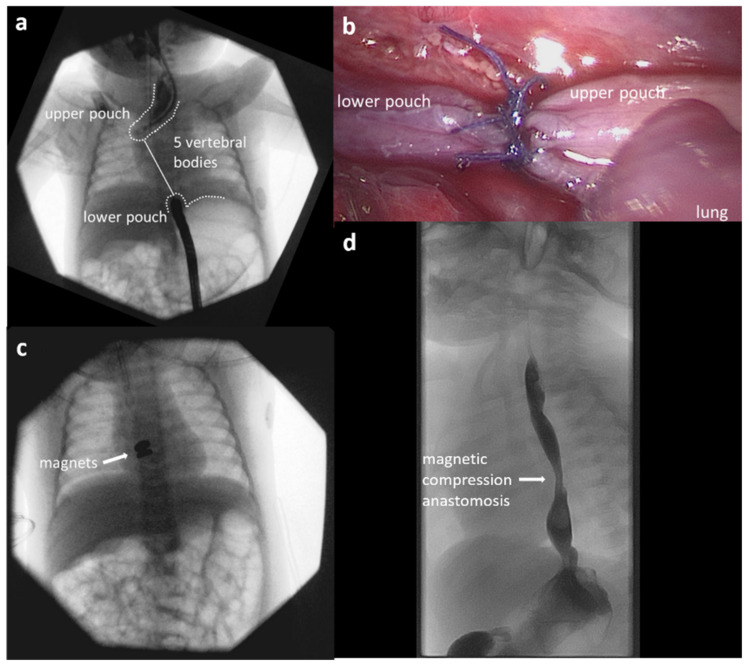
Case report. (**a**) Gap measurement by contrast study at 6 weeks of age. (**b**) Intraoperative view after approximation of the pouches with slip-knot sutures. (**c**) Mated magnets after endoscopic placement. (**d**) Contrast esophagography demonstrating a patent and watertight anastomosis.

**Table 1 children-09-01113-t001:** Inclusion criteria.

Language	English
Date	Any
Subject	Human studies
Study type	Retrospective Prospective Case reports Congress abstracts
Excluded	Videos
Search terms	Magnets Anastomosis Esophagus

**Table 2 children-09-01113-t002:** Overview of available studies for magnetic esophageal compression anastomosis.

Author	Year of Publication	Number of Patients	Type of Esophageal Atresia	Age at Intervention (Mean/Days (d), Months (mo))	Time to Patency (Days)	Anastomotic Leakage
Zaritky et al. [18]	2009	5	Type A: 40% Type B: 40% Type C: 20%	range 30–120 d	4.8 (range 2–7)	none
Zaritky et al. [7]	2014	9	Type A: 66.7% Type C: 33.4%	3 mo (range 23 d–5 mo)	4.2 (range 3–6)	none
Slater et al. [8]	2019	13	Type A: 85% Type C: 15%	4.5 mo (range 2–7.5 mo)	6.3 (range 3–13)	none
Wolfe et al. [12]	2020	1	Type A	n.d.	5	none
Lovvorn et al. [13]	2014	2	Type A: 100%	5.25 mo (range 4–6.5 mo)	7.5 (range 6–10)	none
Dorman et al. [11]	2016	1	Type B	7 mo	13	none
Ellebaek et al. [10]	2018	1	Type A	2.04 mo	5	none
Liu et al. [9]	2020	1	Type B	n.d.	36	1
Conforti et al. [19]	2021	5	Type A: 100%	2.66 mo (range 1.25–3.91 mo)	8 (range 7–9)	none
Muensterer et al. [15]	2021	3	Type A: 33.4% Type B: 33.4% Type C: 33.4%	2.34 mo (range 1.68–3.45 mo)	10.33 (range 7–12)	none

n.d. = no data available.

**Table 3 children-09-01113-t003:** Outcomes of magnetic esophageal compression anastomosis.

Author	Anastomotic Stricture	Number of Esophageal Dilatations (Mean)	Stent Placement	Surgery	Native Esophagus	Follow-Up (Years (y), Months (mo))	Historic Group (HG)/Novel Group (NG)
Zaritky et al. [18]	4/5	n.d.	n.d.	1/5	5/5	n.d.	overlapping patients
Zaritky et al. [7]	8/9	n.d.	2/9	1/9	9/9	9.3 y (range: 0.75–17.75)	overlapping patients
Slater et al. [8]	13/13	9.8 (range 3–22)	6/13	2/13	13/13	n.d.	HG
Wolfe et al. [12]	1/1	13.5 (+/−2.1 SD)	n.d.	n.d.	1/1	11.38 mo (range 14.75–8)	HG
Lovvorn et al. [13]	2/2	3.5 (range 3–4)	none	none	2/2	n.d.	HG
Dorman et al. [11]	1/1	serial dilatation every 2 weeks	n.d.	n.d.	1/1	15 mo	HG
Ellebaek et al. [10]	1/1	17	none	none	1/1	15 mo	HG
Liu et al. [9]	0/1	none	none	none	1/1	18 mo	HG
Conforti et al. [19]	4/5 *	4 (range 3–6)	n.d.	n.d.	5/5	short term	short follow up
Muensterer et al. [15]	3/3	4.33 (range 4–5)	none	none	3/3	15.67 mo (range 14–18)	NG

* Stricture was defined as need for >3 dilatations; n.d. = no data available.

**Table 4 children-09-01113-t004:** Subgroup analysis.

	Historic Group *n* = 19	Novel Device Group *n* = 4	*p*-Value
**Type of esophageal atresia**			
Type A	15 (78.9%)	2 (50%)	
Type B	2 (10.5%)	1 (25%)	0.488
Type C	2 (10.5%)	1 (25%)	
**Interventions before magnetic compression anastomosis**			
Gastrostomy	18 (94.7%)	4 (100%)	0.639
Fistula ligation	3 (15.8%)	2 (50%)	0.132
Approximation	4 (21.1%)	4 (100%)	0.003
**Age at intervention, months (mean +/− SD)**	4.63 +/− 1.71	2.4 +/− 0.8	0.009
**Time to patency, days** **(mean +/− SD)**	8.37 +/− 7.40	9.50 +/− 2.89	0.186
**Complications**			
Leakage	1 (5.3%)	0	0.639
Stenosis	18 (94.7%)	4 (100%)	0.639
**Number of dilatations** **(mean +/− SD)**	9.50 +/− 6.41	4.25 +/− 0.50	0.051
**Stent placement**	6 (35.3%)	0	0.160
**Surgery**	2 (10.5%)	0	0.497
**Maintenance of native esophagus**	19 (100%)	4 (100%)	
**Duration of follow up, years (mean +/− SD)**	6.89 +/− 6.24	1.16 +/− 0.33	0.066

## Data Availability

All data on which this publication is based are available from the corresponding author (OM) upon reasonable request.

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
