# Peer review of "Esophageal Magnetic Compression Anastomosis in Esophageal Atresia Repair: A PRISMA-Compliant Systematic Review and Comparison with a Novel Approach"

_children, 2022, doi:10.3390/children9081113_

Round 1
Reviewer 1 Report
This study aims to assess the outcomes of long gap esophageal atresia treated with magnetic compression anastomosis after thoracoscopic approximation of the pouches. The results of four cases are compared with previously published cases in the literature.
In general, the paper is clearly and well written, and methodologically sound. I think that the paper is worth publishing and I only have a couple of comments:
As the authors have stated in the discussion, the need for dilatations is usually highest during the first months after anastomosis. For more reliable comparison, however, I suggest analyzing ‘dilatations required during the first year after surgery’ or ‘dilatations required during the first year of life’, as the follow-up time was much longer in the historic group. This likely does not change the results significantly, but allows more equal comparison between the groups.
The results are mainly presented as mean and range with one exception of mean and SD. Did the authors consider using median instead of mean due to the small number of patients.
Typo in Table 3 – bottom right cell, should read NG instead of ND, I think.
Author Response
Thank you very much for your careful review.
- As the authors have stated in the discussion, the need for dilatations is usually highest during the first months after anastomosis. For more reliable comparison, however, I suggest analyzing ‘dilatations required during the first year after surgery’ or ‘dilatations required during the first year of life’, as the follow-up time was much longer in the historic group. This likely does not change the results significantly, but allows more equal comparison between the groups.
We agree that comparing dilatations required during the first year after surgery would allow a more equal comparison between the groups. Unfortunately, these detailed data has not been published for patients from the historic group and is therefore not available for the analysis.
- The results are mainly presented as mean and range with one exception of mean and SD. Did the authors consider using median instead of mean due to the small number of patients.
The presentation of the results as mean and range and mean and SD is also due to the presentation of data in the cited publications and therefore cannot be changed.
- Typo in Table 3 – bottom right cell, should read NG instead of ND, I think.
Thank you very much for the comment on the typo in Table 3 – we will correct this.
Reviewer 2 Report
The paper is of extreme interest and addresses a topic that has become central to the discussion of the management of long-gap esophageal atresia in recent years. The paper is well presented. I would like to emphasize two aspects:
1-The description of the magnet used is accurate, but it would help to add pictures.
2-The analytical work comparing a group of patients treated with this new method and a group represented by the results of a systematic review should be carefully considered. In fact, these are two heterogeneous groups both in number and origin of patients. In the HG group, patients come from different studies. This could represent a source of bias.
Author Response
Thank you for your comments and work to improve our manuscript.
1-The description of the magnet used is accurate, but it would help to add pictures.
We added a picture of a pair of magnets (figure 2) to further illustrate the design.
2-The analytical work comparing a group of patients treated with this new method and a group represented by the results of a systematic review should be carefully considered. In fact, these are two heterogeneous groups both in number and origin of patients. In the HG group, patients come from different studies. This could represent a source of bias.
Thank you for your comment. Our aim was to present a systematic review of the available literature and to compare both techniques. For sure, this comparison is limited by the available data and is not comparable to a randomized controlled study. However, we think that the analysis adds important information to this new technique. To further clear the limitations we added the following in line 262 – 264: „We acknowledge that the study groups are heterogeneous, and that the patients in the HG group came from different studies. This could represent a potential source of bias. “
Reviewer 3 Report
That's an interesting study in a very interesting subject.
The study methodology does not seems to be the most appropriated. The authors tried to compare which is very different: There are much more Type A cases in the HG than in the NG and that could be of paramount important. That is even more important because the gap distance on both groups is not available. Was it different? Are the authors comparing "long gap cases" with "less long gap cases"? That would be crucial for any conclusion!
The groups are small and the NG group is just of 4 cases. No comparison between them can be robust enough with such small numbers.
On the NG group the procedure is different from the HG: thoracoscopic esophageal lengthening procedures were performed prior to the magnetic compression anastomosis in all patients on the NG group. That can explain the differences on the outcome and not the magnets used.
Some of the most important outcomes, as stricture formation and time of patency of the anastomosis are similar among the 2 groups. But the time of follow-up is different (shorter on the NG). What if we would wait time enough? Were the NG have more strictures?
As a conclusion, I think this paper is interesting as it adds new insights at the subject but should not be a written as it is actually because the groups are not comparable.
I would suggest the authors to re-write it as a series of 4 patients where a new approach was performed with good initial results: thoracoscopic esophageal lengthening procedures were performed prior to the magnetic compression anastomosis.
Author Response
Thank you for your careful review and comments.
- The study methodology does not seems to be the most appropriated. The authors tried to compare which is very different: There are much more Type A cases in the HG than in the NG and that could be of paramount important. That is even more important because the gap distance on both groups is not available. Was it different? Are the authors comparing "long gap cases" with "less long gap cases"? That would be crucial for any conclusion!
There are 50% of Type A cases in the NG and 78.8% in the HG group. As we mentioned in the methods part information about gap distance was either not available or nonuniform. Unfortunately, there is no standard definition of long-gap. (Line 125-129, “Information about gap distance was nonuniform - either in centimeters or number of vertebral bodies – and was measured at different ages. Therefore, this parameter could not be included in the further analysis.”).
Additionally, we have added the following paragraph to further address this topic (line 260 – 262): “Unfortunately, we only have limited information on the gap length in the HG. This also makes the groups difficult to compare.”
- The groups are small and the NG group is just of 4 cases. No comparison between them can be robust enough with such small numbers
To emphasize this limitation we inserted the following sentence in the limitations paragraph: “Furthermore the group size differed, the NG group is just of 4 cases whereas the HG group includes 19 cases.” (line 264-265).
- On the NG group the procedure is different from the HG: thoracoscopic esophageal lengthening procedures were performed prior to the magnetic compression anastomosis in all patients on the NG group. That can explain the differences on the outcome and not the magnets used.
To answer this point we would like to refer to the discussion (line 219-223; “Our hypothesis is that less anastomotic tension reduces the rate of anastomotic strictures and consequently the number of dilatations, and that the novel curvature and geometry allows for better bridging of the mucosa. Nevertheless, almost all patients in both groups developed some esophageal stenosis...”). From our perspective, the trend to lower rates of dilatations may be explained by both alterations: the new magnet design and the lengthening procedure. To further address this point we added the following in line 215-218: “It is important to understand that magnets should not be used for approximation, but only for anastomosis. Therefore, performing a thoracoscopic approximation before placing the magnets is part of the comprehensive treatment strategy in the NG.”
Round 2
Reviewer 3 Report
- The study methodology does not seems to be the most appropriated. The authors tried to compare which is very different: There are much more Type A cases in the HG than in the NG and that could be of paramount important. It did not reach a p-value that would suggest a statistical significance just because the numbers are very small. That is even more important because the gap distance on both groups is not available. So it is nod adequate to compare these two groups.
- On the NG group the procedure is different from the HG: thoracoscopic esophageal lengthening procedures were performed prior to the magnetic compression anastomosis in all patients on the NG group. That can explain the differences on the outcome and not the magnets used. So, as added now, this paper does not provide an explanation for the outcome: is is due to the lengthening procedure, the new magnets used or both? And again, it is not possible to compare with the other group due to their differences.
- So, I think that a case series (4 cases) would be the best way to present these impressions, since it is not adequate to compare it with the historical group as proposed.
Author Response
- The study methodology does not seems to be the most appropriated. The authors tried to compare which is very different: There are much more Type A cases in the HG than in the NG and that could be of paramount important. It did not reach a p-value that would suggest a statistical significance just because the numbers are very small. That is even more important because the gap distance on both groups is not available. So it is nod adequate to compare these two groups.
In our article, we clearly and transparently present all data, so that the reader can judge by themself if they find the comparison of a current to a historic series appropriate. This article is primarily a PRISMA compliant systematic review, the comparison is to an emerging, novel approach. Since this is a rare disease, low numbers are to be expected. However, this does not preclude a comparison, as long as the data involved is presented in a transparent fashion.
2. On the NG group the procedure is different from the HG: thoracoscopic esophageal lengthening procedures were performed prior to the magnetic compression anastomosis in all patients on the NG group. That can explain the differences on the outcome and not the magnets used. So, as added now, this paper does not provide an explanation for the outcome: is is due to the lengthening procedure, the new magnets used or both? And again, it is not possible to compare with the other group due to their differences.
The novel concept comprises previous approximation of the pouches, separating the tension operation from the anastomosis. The novel method is not simply a different magnet, but a treatment concept that entails previous thoracoscopic approximation and later anastomosis using the magnets. We believe that this is beneficial, because it potentially lowers the need for dilatations. All this has been clearly stated in the discussion section.
3. So, I think that a case series (4 cases) would be the best way to present these impressions, since it is not adequate to compare it with the historical group as proposed.
The focus on this paper is not the case series, but the PRISMA Compliant Systematic Review. We have already published a case series of the first three cases [Muensterer OJ et al. Novel Device for Endoluminal Esophageal Atresia Repair: First-in-Human Experience. Pediatrics. 2021; 148:e2020049627], but the point of this article is the systematic review of the literature published on the 2 methods so far, along with the one previously not published case.
As mentioned above, the focus on this paper is not the case series but the PRISMA Compliant Systematic Review.